# Hedging in games: Faster convergence of external and swap regrets

**Xi Chen**
Department of Computer Science
Columbia University
xichen@cs.columbia.edu

**Binghui Peng**
Department of Computer Science
Columbia University
bp2601@columbia.edu

## Abstract

We consider the setting where players run the Hedge algorithm or its optimistic variant to play an $n$-action game repeatedly for $T$ rounds.

- For two-player games, we show that the regret of optimistic Hedge decays at rate $O(1/T^{5/6})$, improving the previous bound of $O(1/T^{3/4})$ by Syrgkanis, Agarwal, Luo and Schapire [27].

- In contrast, we show that the convergence rate of vanilla Hedge is no better than $O(1/\sqrt{T})$, addressing an open question posed in Syrgkanis, Agarwal, Luo and Schapire [27].

For general $m$-player games, we show that the swap regret of each player decays at $O(m^{1/2}(n \log n/T)^{3/4})$ when they combine optimistic Hedge with the classical external-to-internal reduction of Blum and Mansour [6]. Via standard connections, our new (swap) regret bounds imply faster convergence to coarse correlated equilibria in two-player games and to correlated equilibria in multiplayer games.

## 1 Introduction

Online algorithms for regret minimization play an important role in many applications in machine learning where real-time sequential decision making is crucial [19, 7, 26]. A number of algorithms have been developed, including Hedge / Multiplicative Weights [2], Mirror Decent [19], Follow the Regularized / Perturbed Leader [20], and their power and limits against an adversarial environment have been well understood: The average (external) regret decays at a rate of $O(1/\sqrt{T})$ after $T$ rounds, which is known to be tight for any online algorithm.

What happens if players in a repeated game run one of these algorithms? Given that they are now running against similar algorithms over a fixed game, could the regret of each player decay significantly faster than $1/\sqrt{T}$? This was answered positively in a sequence of works [9, 24, 27]. Among these results, the one that is most relevant to ours is that of Syrgkanis, Agarwal, Luo and Schapire [27]. They showed that if every player in a multiplayer game runs an algorithm that satisfies the RVU (Regret bounded by Variation in Utilities) property, then the regret of each player decays at $O(1/T^{3/4})$. *Can this bound be further improved?*

Besides regret minimization, understanding no-regret dynamics in games is motivated by connections with various equilibrium concepts [15, 13, 12, 18, 6, 17, 22]. For example, if every player runs an algorithm with vanishing regret, then the empirical distribution must converge to a *coarse* correlated equilibrium [7]. Nevertheless, to converge to a more preferred correlated equilibrium [3], a stronger notion of regrets called *swap regrets* (see Section 2) is required [13, 18, 6]. The minimization of swap regrets under the adversarial setting was studied by Blum and Mansour [6]. They gave a generic reduction from regret minimization algorithms which led to a tight $O(\sqrt{n \log n/T})$-bound for the

average swap regret. A natural question is *whether a speedup similar to that of* [27] *is possible for swap regrets in the repeated game setting.*

**Our contributions: Faster convergence of swap regrets.** We give the first algorithm that achieves an average swap regret that is significantly lower than $O(1/\sqrt{T})$ under the repeated game setting. This algorithm, denoted by `BM-Optimistic-Hedge`, combines the external-to-internal reduction of [6] with the optimistic Hedge algorithm [24, 27] as its regret minimization component. (Optimistic Hedge can be viewed as an instantiation of the optimistic Follow the Regularized Leader algorithm; see Section 2.) We show that if every player in a repeated game of $m$ players and $n$ actions runs `BM-Optimistic-Hedge`, then the average swap regret is at most $O(m^{1/2}(n \log n/T)^{3/4})$; see Theorem 5.1 in Section 5. Via the relationship between correlated equilibria and swap regrets, our result implies faster convergence to a correlated equilibrium. When specialized to two-player games, the empirical distribution of players running `BM-Optimistic-Hedge` converges to an $\epsilon$–correlated equilibrium after $O(n \log n/\epsilon^{4/3})$ rounds, improving the $O(n \log n/\epsilon^2)$ bound of [6].

Our main technical lemma behind Theorem 5.1 shows that strategies produced by the algorithm of [6] with optimistic Hedge moves very slowly in $\ell_1$-norm under the adversarial setting (which in turn allows us to apply a stability argument similar to [27]). This came as a surprise because a key component of the algorithm of [6] each round is to compute the stationary distribution of a Markov chain, which is highly sensitive to small changes in the Markov chain. We overcome this difficulty by exploiting the fact that Hedge only incurs small *multiplicative* changes to the Markov chain, which allows us to bound the change in the stationary distribution using the classical Markov chain tree theorem. As a consequence, our algorithm enjoys the benefit of faster convergence when playing with each other, while remain robust against adversaries.

**Our contributions: Hedge in two-player games.** In addition we consider regret minimization in a two-player game with $n$ actions using either vanilla or optimistic Hedge. We show that optimistic Hedge can achieve an average regret of $O(1/T^{5/6})$, improving the bound $O(1/T^{3/4})$ by [27] for two-player games; see Theorem 3.1 in Section 3. In contrast, we show that even under this game-theoretic setting, vanilla Hedge cannot asymptotically outperform the $O(1/\sqrt{T})$ adversarial bound; see Theorem 4.1 in Section 4. This addresses an open question posed by [27] concerning the convergence rate of vanilla Hedge in a repeated game.

The key step in our analysis of optimistic Hedge is to show that, even under the adversarial setting, the trajectory length of strategy movements (in their squared $\ell_1$-norm) can be bounded using that of cost vectors (in $\ell_\infty$-norm); see Lemma 3.2. (Intuitively, it is unlikely for the strategy of optimistic Hedge to change significantly over time while the loss vector stays stable.) This allows us to build a strong relationship between the trajectory length of each player's strategy movements, and then use the RVU property of optimistic Hedge to bound their individual regrets.

Our lower bounds for vanilla Hedge use three very simple $2 \times 2$ games to handle different ranges of the learning rate $\eta$. For the most intriguing case when $\eta$ is at least $\Omega(1/\sqrt{n})$ and bounded from above by some constant, we study the zero-sum Matching Pennies game and use it to show that the overall regret of at least one player is $\Omega(\sqrt{T})$. Our analysis is inspired by the result of [5] which shows that the KL divergence of strategies played by Hedge in a two-player zero-sum game is strictly increasing. For Matching Pennies, we start with a quantitative bound on how fast the KL divergence grows in Lemma 4.3. This implies the existence of a window of length $\sqrt{T}$ during which the cost of one of the player grows by $\Omega(1)$ each round; the zero-sum structure of the game allows us to conclude that at least one of the players must have regret at least $\Omega(\sqrt{T})$ at some point in this window.

## 1.1 Related work

Initiated by Daskalakis, Deckelbaum and Kim [9], there has been a sequence of works that study no-regret learning algorithms in games [24, 27, 14, 29]. Daskalakis et al. [9] designed an algorithm by adapting Nesterov's accelerated saddle point algorithm to two-player *zero-sum games*, and showed that if both players run this algorithm then their average regrets decay at rate $O(1/T)$, which is optimal. Later Rakhlin and Sridharan [23, 24] developed a simple and intuitive family of algorithms, i.e. *optimistic Mirror Descent* and *optimistic Follow the Regularized Leader*, that incorporate predictions into the strategy. They proved that if both players adopt the algorithm, then their average regrets also decay at rate $O(1/T)$ in *zero sum games*. Syrgkanis et al. [27] further strengthened this line of works by showing that in a *general $m$-player game*, if every player runs an algorithm

that satisfies the RVU property then the average regret decays at rate $O(1/T^{3/4})$. Syrgkanis et al. [27] also considered the convergence of social welfare and proved an even faster rate of $O(1/T)$ in smooth games [25]. Foster et al. [14] extended [27] and showed that if one only aims for an approximately optimal social welfare, then the class of algorithms allowed can be much broader. Recently, Daskalakis and Panageas [11] proved the last iteration convergence of optimistic Hedge in zero-sum game, i.e., instead of averaging over the trajectory, they showed that optimistic Hedge converges to a Nash equilibrium in a zero-sum game.

There is also a growing body of works [21, 5, 4, 8] on the dynamics of no-regret learning over games in the last few years. Most of these works studied the dynamics of no-regret learning from a dynamical system point of view and provided qualitative intuition on the evolution of no-regret learning. Among them, [4] is most relevant, in which Bailey and Piliouras proved an $\Omega(\sqrt{T})$ lower bound on the convergence rate of online gradient descent [30] for the $2 \times 2$ Matching Pennies game. However, we remark that their lower bound only works for online gradient descent and they need to fix the learning rate $\eta$ to 1. Our lower bound for vanilla Hedge in two-player games holds for arbitrary learning rates.

## 2 Preliminary

**Notation.** Given two positive integers $n \leq m$, we use $[n]$ to denote $\{1, \ldots, n\}$ and $[n : m]$ to denote $\{n, \ldots, m\}$. We use $D_{\mathrm{KL}}(p\|q)$ to denote the KL divergence with natural logarithm.

**Repeated games and regrets.** Consider a game $G$ played between $m$ players, where each player $i \in [m]$ has a strategy space $S_i$ with $|S_i| = n$ and a *loss* function $\mathcal{L}_i : S_1 \times \cdots \times S_m \to [0, 1]$ such that $\mathcal{L}_i(\mathbf{s})$ is the loss of player $i$ for each pure strategy profile $\mathbf{s} = (s_1, \ldots, s_n) \in S_1 \times \cdots \times S_m$. A mixed strategy for player $i$ is a probability distribution $x_i$ over $S_i$, where the $j$th action is played with probability $x_i(j)$. Given a mixed (or pure) strategy profile $\mathbf{x} = (x_1, \ldots, x_m)$ (or $\mathbf{s} = (s_1, \ldots, s_m)$), we write $\mathbf{x}_{-i}$ (or $\mathbf{s}_{-i}$) to denote the profile after removing $x_i$ (or $s_i$, respectively).

We consider the scenario where the $m$ players play $G$ repeatedly for $T$ rounds. At the beginning of each round $t$, $t \in [T]$, each player $i$ picks a mixed strategy $x_i^t$ and let $\mathbf{x}^t = (x_1^t, \ldots, x_m^t)$ be the mixed strategy profile. We consider the *full information* setting where each player observes the *expected* loss of *all* her actions. Formally, player $i$ observes a loss vector $\ell_i^t$ with $\ell_i^t(j) = \mathbb{E}_{\mathbf{s}_{-i} \sim \mathbf{x}_{-i}^t}[\mathcal{L}_i(j, \mathbf{s}_{-i})]$, and her expected loss is given by $\langle x_i^t, \ell_i^t \rangle$. At the end of round $T$, the *regret* of player $i$ is

$$\mathrm{regret}_T^i = \sum_{t \in [T]} \langle x_i^t, \ell_i^t \rangle - \min_{j \in [n]} \sum_{t \in [T]} \ell_i^t(j), \tag{1}$$

i.e., the maximum gain one could have obtained by switching to some fixed action. A stronger notion of regret, referred as *swap regret*, is defined as

$$\mathrm{swap\text{-}regret}_T^i = \sum_{t \in [T]} \langle x_i^t, \ell_i^t \rangle - \min_\phi \sum_{t \in [T]} \sum_{j \in [n]} x_i^t(j) \cdot \ell_i^t(\phi(j)), \tag{2}$$

where the minimum is over all $n^n$ (swap) functions $\phi : [n] \to [n]$ that swap action $j$ with $\phi(j)$. The swap regret equals the maximum gain one could have achieved by using a fixed swap function over its past mixed strategies.

**Hedge.** Consider the adversarial online model where a player has $n$ actions and picks a distribution $x^t$ over them at the beginning of each round $t$. During round $t$ the player receives a loss vector $\ell^t$ and pays a loss of $\langle x^t, \ell^t \rangle$. The vanilla Hedge algorithm [16] with learning rate $\eta > 0$ starts by setting $x^1$ to be the uniform distribution and then keeps applying the following updating rule to obtain $x^{t+1}$ from $x^t$ and the loss vector $\ell^t$ at the end of round $t$: for each action $j \in [n]$,

$$x^{t+1}(j) = \frac{x^t(j) \cdot \exp(-\eta \cdot \ell^t(j))}{\sum_{k \in [n]} x^t(k) \cdot \exp(-\eta \cdot \ell^t(k))}.$$

On the other hand, the optimistic Hedge algorithm can be obtained from the *optimistic follow the regularized leader* proposed by [24, 27], and have the following updating rule:

$$x^{t+1}(j) = \frac{x^t(j) \cdot \exp(-\eta(2\ell^t(j) - \ell^{t-1}(j))}{\sum_{k \in [n]} x^t(k) \cdot \exp(-\eta(2\ell^t(k) - \ell^{t-1}(k)))}, \tag{3}$$

with $\ell^0 = \mathbf{0}$ being the all-zero vector. We have the following regret bound for optimistic Hedge.

**Lemma 2.1** ([24, 27]). *Under the adversarial setting, optimistic Hedge satisfies*

$$\text{regret}_T \leq \frac{2\log n}{\eta} + \eta \sum_{t\in[T]} \|\ell^t - \ell^{t-1}\|_\infty^2 - \frac{1}{4\eta}\sum_{t\in[T]} \|x^{t+1} - x^t\|_1^2. \tag{4}$$

## 3 Optimistic Hedge in Two-Player Games

In this section we analyze the performance of the optimistic Hedge algorithm when it is used by two players to play a (general, not necessarily zero-sum) $n \times n$ game repeatedly.

**Theorem 3.1.** *Suppose both players in a two-player game run optimistic Hedge for $T$ rounds with learning rate $\eta = (\log n/T)^{1/6}$. Then the individual regret of each player is $O(T^{1/6}\log^{5/6} n)$.*

We assume without loss of generality that $T \geq \log n$; otherwise, the regret of each player is trivially at most $T \leq T^{1/6}\log^{5/6} n$. The following lemma is essential to our proof of Theorem 3.1. Consider the adversarial online setting where a player runs optimistic Hedge for $T$ rounds. The lemma bounds the trajectory length of the strategy movement using that of cost vectors.

**Lemma 3.2.** *Suppose that a player runs optimistic Hedge with learning rate $\eta$ for $T$ rounds. Let $\ell^0, \ell^1, \ldots, \ell^T$ be the cost vectors with $\ell^0 = \mathbf{0}$ and $x^1, \ldots, x^T$ be the strategies played. Then*

$$\sum_{t\in[2:T]} \|x^t - x^{t-1}\|_1^2 \leq O(\log n) + O(\eta + \eta^2)\sum_{t\in[T-1]} \|\ell^t - \ell^{t-1}\|_\infty. \tag{5}$$

*Proof of Theorem 3.1 assuming Lemma 3.2.* Let $G = (A, B)$ be the game, where $A, B \in [0,1]^{n\times n}$ denote the cost matrices of the first and second players, respectively. We use $x^t$ and $y^t$ to denote strategies played by the two players and use $\ell_x^t$ and $\ell_y^t$ to denote their cost vectors in the $t$th round. So we have $\ell_x^t = Ay^t$ and $\ell_y^t = B^T x^t$. Therefore, we have for each $t \geq 2$:

$$\|\ell_y^t - \ell_y^{t-1}\|_\infty = \|B^T(x^t - x^{t-1})\|_\infty \leq \|x^t - x^{t-1}\|_1 \quad \text{and} \tag{6}$$

$$\|\ell_x^t - \ell_x^{t-1}\|_\infty = \|A(y^t - y^{t-1})\|_\infty \leq \|y^t - y^{t-1}\|_1.$$

Without loss of generality it suffices to bound the regret of the second player. Set $\eta = (\log n/T)^{1/6}$ with $T \geq \log n$ so that $\eta \leq 1$. We have

$$\text{regret}_T^y \leq \frac{2\log n}{\eta} + \eta \sum_{t\in[T]} \|\ell_y^t - \ell_y^{t-1}\|_\infty^2 - \frac{1}{4\eta}\sum_{t\in[T]} \|y^{t+1} - y^t\|_1^2 \qquad \text{Lemma 2.1}$$

$$\leq \frac{2\log n}{\eta} + \eta + \eta\sum_{t\in[2:T]} \|x^t - x^{t-1}\|_1^2 - \frac{1}{4\eta}\sum_{t\in[2:T+1]} \|\ell_x^t - \ell_x^{t-1}\|_\infty^2 \qquad \text{using (6)}$$

$$\leq \frac{2\log n}{\eta} + \eta + \eta\left(O(\log n) + O(\eta)\sum_{t\in[T-1]} \|\ell_x^t - \ell_x^{t-1}\|_\infty\right)$$

$$\quad - \frac{1}{4\eta}\sum_{t\in[T-1]} \|\ell_x^t - \ell_x^{t-1}\|_\infty^2 + \frac{1}{4\eta} \qquad \text{Lemma 3.2}$$

$$= O\left(\frac{\log n}{\eta}\right) + \sum_{t\in[T-1]}\left(O(\eta^2)\cdot\|\ell_x^t - \ell_x^{t-1}\|_\infty - \frac{1}{4\eta}\cdot\|\ell_x^t - \ell_x^{t-1}\|_\infty^2\right)$$

$$\leq O\left(\frac{\log n}{\eta}\right) + T\cdot O(\eta^5) = O\left(T^{1/6}\log^{5/6} n\right).$$

This finishes the proof of the theorem. $\qquad\qquad\square$

## 4 Lower Bounds for Hedge in Two-Player Games

We prove lower bounds for regrets of players when they both run the vanilla Hedge algorithm. We show that even in games with two actions, vanilla Hedge cannot perform asymptotically better than its guaranteed regret bound of $O(\sqrt{T})$ under the adversarial setting.

**Theorem 4.1.** *Suppose two players run the vanilla Hedge algorithm to play a two-action game with initial strategy $(0.4, 0.6)$. Then for any sufficiently large $T$ and any learning rate $\eta > 0$, there is a game such that at least one player has regret $\Omega(\sqrt{T})$ after $T'$ rounds for some $T' \in [T : T + \sqrt{T}]$.*

**Remark 4.2.** *Theorem 4.1 shows that even if players have a good estimation about the number of rounds to play (i.e., between $T$ and $T + \sqrt{T}$), vanilla Hedge with any learning rate $\eta(T) > 0$ picked using $T$ cannot promise to achieve a regret bound that is asymptotically lower than $O(\sqrt{T})$ for every round $T' \in [T : T + \sqrt{T}]$. We would like to point out that the use of $(0.4, 0.6)$ as the initial strategy instead of the uniform distribution is not crucial but only to simplify the construction and analysis.*

Let $T$ be a sufficiently large integer. We will use three games $G_i = (A, B_i)$, $i \in \{1, 2, 3\}$, to handle three cases of the learning rate $\eta$, where

$$ A = \begin{pmatrix} 1 & -1 \\ -1 & 1 \end{pmatrix}, \quad B_1 = \begin{pmatrix} -1 & 1 \\ 1 & -1 \end{pmatrix}, \quad B_2 = \begin{pmatrix} 1 & 1 \\ 1 & 1 \end{pmatrix} \quad \text{and} \quad B_3 = \begin{pmatrix} 1 & -1 \\ -1 & 1 \end{pmatrix}. $$

We use $G_2$ to handle the case when $\eta \leq 64/(c_0\sqrt{T})$ where $c_0 \in (0, 1]$ is a constant introduced below in Lemma 4.3. We use $G_3$ to handle the case when $\eta \geq 3$. The most intriguing case is when the learning rate $\eta$ is between $64/(c_0\sqrt{T})$ and $3$. For this case we use the Matching Pennies game $G_1 = (A, B_1)$.

Let $x^t$ and $y^t$ denote strategies played in round $t$ by the first and second players, respectively. Let $x^\star = y^\star = (0.5, 0.5)$. The proof for this case relies on the following lemma, which shows that the KL divergence between $(x^\star, y^\star)$ and $(x^T, y^T)$ after $T$ rounds is at least $\Omega(\sqrt{T}\eta)$).

**Lemma 4.3.** *Suppose players run vanilla Hedge for $T$ rounds with $\eta : 16/\sqrt{T} \leq \eta \leq 3$. Then*

$$ D_{KL}(x^\star \| x^T) + D_{KL}(y^\star \| y^T) \geq c_0\sqrt{T}\eta, \quad \text{for some constant } c_0 \in (0, 1]. $$

We are now ready to prove Theorem 4.1 for the main case when $64/(c_0\sqrt{T}) \leq \eta \leq 3$.

*Proof of Theorem 4.1 for the main case.* For convenience we let $x_t = x^t(1)$ (or $y_t = y^t(1)$) denote the probability of playing the first action in $x^t$ (or $y^t$, respectively). We first describe the high level idea behind the proof. Since we know the KL divergence is at least $c_0\sqrt{T}\eta$ at time $T$ by Lemma 4.3, at least one of $x_T$ and $y_T$ is extremely close to either $0$ or $1$. Assume without loss of generality that this is the case for $x_T$. As a result, the probability of the first player playing the first action will not change much for the next $\sqrt{T}$ rounds. Consequently, during the next $\sqrt{T}$ rounds, one of the players must keep losing and the other player will keep winning. This can be used to show that one of the two players must have regret at least $\Omega(\sqrt{T})$ at some point $T'$ between $T$ and $T + \sqrt{T}$.

To make this more formal, let $\ell_x^t$ (or $\ell_y^t$) denote the cost vector of the first (or the second) player at round $t$ and define $L_x^t$ and $L_y^t$ to be the total loss up to round $t$ of the two players:

$$ L_x^t = \sum_{\tau \in [t]} \langle x^\tau, \ell_x^\tau \rangle \quad \text{and} \quad L_y^t = \sum_{\tau \in [t]} \langle y^\tau, \ell_y^\tau \rangle. $$

Since $G_1 = (A, B_1)$ is zero-sum, we have $\langle x^\tau, \ell_x^\tau \rangle + \langle y^\tau, \ell_y^\tau \rangle = 0$ and thus, $L_x^t + L_y^t = 0$. Moreover, noting that the sum of two rows of $A$ is zero, the first player can always guarantee an overall loss of at most $0$ when playing the best fixed action in hindsight. Therefore, $\text{regret}_t^x \geq L_x^t$ and similarly $\text{regret}_t^y \geq L_y^t$. Combining this with $L_x^t + L_y^t = 0$, we have

$$ \max\left\{ \text{regret}_t^x, \text{regret}_t^y \right\} \geq |L_x^t| = |L_y^t|. $$

To finish the proof, it suffices to show that

$$ \left|L_x^{T'}\right| = \left|L_y^{T'}\right| \geq \Omega(\sqrt{T}), \quad \text{for some } T' \in [T : T + \sqrt{T}]. \tag{7} $$

Let $L = c_0\sqrt{T}/8 \leq \sqrt{T}$. We have from Lemma 4.3 that the KL divergence is at least $c_0\sqrt{T}\eta$ (using $\eta \geq 64/(c_0\sqrt{T}) > 16/\sqrt{T}$). We assume without loss of generality that $D_{\text{KL}}(x^\star \| x^T) \geq c_0\sqrt{T}\eta/2$. We further assume without loss of generality that the second term is larger:

$$ \frac{1}{2} \cdot \log \frac{1}{2(1 - x_T)} \geq \frac{c_0\sqrt{T}\eta}{4}. $$

It follows that $x_T$ is very close to 1: $x_T \geq 1 - \exp(-c_0\sqrt{T}\eta/2)$, and we use this to show that $x_{T+\tau}$ remains close to 1 for all $\tau \in [L]$. To see this is the case, we note that

$$\frac{x_{T+\tau}}{1 - x_{T+\tau}} \geq \exp(-2\eta\tau) \cdot \frac{x_T}{1 - x_T} \geq \frac{1}{2} \cdot \exp\left(-2\eta L + \frac{c_0\sqrt{T}\eta}{2}\right) = \frac{1}{2} \cdot \exp\left(\frac{c_0\sqrt{T}\eta}{4}\right) \geq 3,$$

where we used $\eta \geq 64/(c_0\sqrt{T})$ in the last inequality. This implies $x_{T+\tau} \geq 3/4$ for all $\tau \in [L]$.

Now we turn our attention to the second player. Given that $x_{T+\tau} \geq 3/4$ for all $\tau \in [L]$, $y_{T+\tau}$ keeps growing for all $\tau \in [L]$. As a result there is an interval $I \subseteq [L]$ such that (i) every $y_{T+\tau}$, $\tau \in I$, lies between $1/4$ and $3/4$; (ii) every $y_{T+\tau}$ before $I$ is smaller than $1/4$; and (iii) every $y_{T+\tau}$ after $I$ is larger than $3/4$. Using a similar argument, we show that $I$ cannot be too long. Letting $\ell$ and $r$ be the left and right endpoints of $I$, we have

$$3 \geq \frac{y_r}{1 - y_r} \geq \exp\left(\frac{\eta(r - \ell)}{2}\right) \cdot \frac{y_\ell}{1 - y_\ell} \geq \exp\left(\frac{\eta(r - \ell)}{2}\right) \cdot \frac{1}{3}.$$

As a result, we have $(r - \ell) \leq 6/\eta \leq (3/32) \cdot c_0\sqrt{T}$ and thus, either (i) or (ii) is of length at least $\Omega(L)$. We focus on the case when (ii) is long; the other case can be handled similarly.

Summarizing what we have so far, there is an interval $J = [\alpha : \beta] \subseteq [L]$ of length $\Omega(L)$ such that for every $\tau \in J$, both $x_{T+\tau}$ and $y_{T+\tau}$ are at least $3/4$. This implies that the total loss of the first player grows by $\Omega(1)$ each round and thus, $L_x^{T+\beta} - L_x^{T+\alpha} \geq \Omega(L)$. Therefore, either $|L_x^{T+\alpha}| \geq \Omega(L)$ or $|L_x^{T+\beta}| \geq \Omega(L)$. This finishes the proof of (7) using $L = \Omega(\sqrt{T})$ and the proof of the theorem. $\quad\square$

## 5  Faster Convergence of Swap Regrets

Under the adversarial online model, Blum and Mansour [6] gave a black-box reduction showing that any algorithm that achieve good regrets can be converted into an algorithm that achieves good swap regrets. In this section we show that if every player in a repeated game runs their algorithm with optimistic Hedge as its core, then the swap regret of each player can be bounded from above by $O((n \log n)^{3/4}(mT)^{1/4})$, where $m$ is the number of players and $n$ is the number of actions.

We start with an overview on the reduction framework of [6], which we will refer to as the BM algorithm. Let $S = [n]$ be the set of available actions. Given an algorithm `ALG` that achieves good regrets, the BM algorithm instantiates $n$ copies $\texttt{ALG}_1, \ldots, \texttt{ALG}_n$ of `ALG` over $S$. At the beginning of each round $t = 1, \ldots, T$, the BM algorithm receives a distribution $q_i^t$ over $S$ from $\texttt{ALG}_i$ for each $i \in [n]$, and plays $x^t$, which is the unique distribution over $S$ that satisfies $x^t = x^t Q^t$, where $Q^t$ is the $n \times n$ matrix with row vectors $q_1^t, \ldots, q_n^t$. After receiving the loss vector $\ell^t$, the BM algorithm experiences a loss of $\langle x^t, \ell^t \rangle$ and distributes $x^t(i) \cdot \ell^t$ to $\texttt{ALG}_i$ as its loss vector in round $t$.

We are now ready to state our main theorem of this section:

**Theorem 5.1.** *Suppose that every player in a repeated game runs the BM algorithm with optimistic Hedge as* `ALG` *and sets the learning rate of the latter to be* $\eta = (n \log n/(m^2 T))^{1/4}$. *Then the swap regret of each player is* $O((n \log n)^{3/4} \cdot (m^2 T)^{1/4})$.

For convenience we refer to the BM algorithm with optimistic Hedge as `BM-Optimistic-Hedge` in the rest of the section. We first combine the analysis of [6] for the BM algorithm and Lemma 3 to obtain the following bound for the swap regret of `BM-Optimistic-Hedge` under the adversarial setting, in terms of the total path length of cost vectors the player's mixed strategies:

**Lemma 5.2.** *Suppose that a player runs* `BM-Optimistic-Hedge` *with* $\eta > 0$ *for* $T$ *rounds. Then*

$$\text{swap-regret}_T \leq \frac{2n \log n}{\eta} + 2\eta \left(\sum_{t=2}^{T} \|x^t - x^{t-1}\|_1^2 + \sum_{t=1}^{T} \|\ell^t - \ell^{t-1}\|_\infty^2\right), \quad \text{where } \ell^0 = \mathbf{0}.$$

The proof can be found in Appendix **??**. For the repeated game setting, we have for each $t \geq 2$,

$$\|\ell_i^t - \ell_i^{t-1}\|_\infty \leq \|\mathbf{x}_{-i}^t - \mathbf{x}_{-i}^{t-1}\|_1 \leq \sum_{j \neq i} \|\mathbf{x}_j^t - \mathbf{x}_j^{t-1}\|_1$$

$$Q = \begin{pmatrix} 1 - \epsilon & \epsilon \\ \epsilon' & 1 - \epsilon' \end{pmatrix} \quad x = \begin{pmatrix} \dfrac{1}{k+1} & \dfrac{k}{k+1} \end{pmatrix} \quad \text{vs} \quad Q = \begin{pmatrix} 1 - \epsilon' & \epsilon' \\ \epsilon & 1 - \epsilon \end{pmatrix} \quad x = \begin{pmatrix} \dfrac{k}{k+1} & \dfrac{1}{k+1} \end{pmatrix}$$

Figure 1: Let $\epsilon' = \epsilon/k$. Additive perturbations may change the stationary distribution dramatically.

where the last inequality used the fact that both $\mathbf{x}^t_{-i}$ and $\mathbf{x}^{t-1}_{-i}$ are product distributions. Combining it with Lemma 5.2, we can bound the swap regret of each player $i \in [m]$ in the game by

$$\text{swap-regret}^i_T \leq \frac{2n \log n}{\eta} + 2\eta + 2\eta m \sum_{j \in [m]} \sum_{t=2}^{T} \|x^t_j - x^{t-1}_j\|^2_1. \tag{8}$$

We prove the following main technical lemma in the rest of the section, which states that the mixed strategy $x^t$ produced by `BM-Optimistic-Hedge` under the adversarial setting moves very slowly (by at most $O(\eta)$ in $\ell_1$-distance each round). Theorem 5.1 follows by combining Lemma 5.2 and 5.3.

**Lemma 5.3.** *Suppose that a player runs* `BM-Optimistic-Hedge` *with rate* $\eta : 0 < \eta \leq 1/6$ *under the adversarial setting. Then we have* $\|x^t - x^{t-1}\|_1 \leq O(\eta)$ *for all* $t \geq 2$.

*Proof of Theorem 5.1 Assuming Lemma 5.3.* Let $\eta = (n \log n)^{1/4}(m^2 T)^{-1/4}$. For the special case when $\eta > 1/6$, the swap regret of each player is trivially at most $T = O((n \log n)^{3/4} \cdot (m^2 T)^{1/4})$. Assuming $\eta \leq 1/6$, by Lemma 5.2 we have from (8) that

$$\text{swap-regret}^i_T \leq \frac{2n \log n}{\eta} + 2\eta + 2\eta m^2 T \cdot O(\eta^2) = O\left((n \log n)^{3/4} \cdot (m^2 T)^{1/4}\right).$$

This finishes the proof of the theorem. $\qquad\qquad\qquad\qquad\qquad\qquad\qquad\qquad\qquad\qquad\square$

The proof of Lemma 5.3 can be found in Appendix. Here we give a high-level description of its proof. Given that `BM-Optimistic-Hedge` runs $n$ copies of optimistic Hedge with rate $\eta$, we know that mixed strategies proposed by each $\text{ALG}_i$ move very slowly: $\|q^t_i - q^{t-1}_i\|_1 \leq O(\eta)$. However, it is not clear whether this translates into a similar property for $x^t$ since the latter is obtained by solving $x^t = x^t Q^t$. Equivalently, $x^t$ can be viewed as the stationary distribution of the Markov chain $Q^t$ composed by strategies of each individual expert $\text{ALG}_i$, and its dependency on $Q^t$ is highly nonlinear. While there is a vast literature on the perturbation analysis of Markov chains, many results require additional assumptions on the underlying Markov chain (e.g. bounded eigenvalue gap) and are not well suited for our setting here. Indeed, it is easy to come up with examples showing that the stationary distrbution is extremely sensitive to small *additive* perturbations (see Figure 1). As a result one cannot hope to prove Lemma 5.3 based on the property $\|q^t_i - q^{t-1}_i\|_1 \leq O(\eta)$ only.

We circumvent this difficulty by noting that optimistic Hedge only incurs small *multiplicative* perturbations on the Markov chain, i.e. each entry of $Q^t$ differs from the corresponding entry of $Q^{t-1}$ by no more than a small multiplicative factor of the latter. We then present an analysis on stationary distributions of Markov chains under multiplicative perturbations, based on the classical Markov chain tree theorem, and then use it to prove Lemma 5.3.

We further prove that one can design a wrapper for `BM-Optimistic-Hedge` that is robust against adversarial opponents:

**Corollary 5.4.** *There is an algorithm* `BM-Optimistic-Hedge`$^*$ *with the following guarantee. If all players run* `BM-Optimistic-Hedge`$^*$, *then the swap regret of each individual is* $\tilde{O}(n^{3/4}(m^2 T)^{1/4})$; *if the player is facing adversaries, then the swap regret is still at most* $\tilde{O}((nT)^{1/2} + n^{3/4}(m^2 T)^{1/4})$.

In the full version we give two more extensions to our results on swap regrets.

1. We show that incorporating optimistic Hedge into a folklore algorithm from [7] can also achieve faster convergence of swap regrets, with a slightly worse dependence on $n$. Interestingly, our analysis of this algorithm also crucially relies on the perturbation analysis of stationary distributions of Markov chains.

2. We study the convergence to the approximately optimal social welfare (following the definition in [14]) with no-swap regret algorithms, and prove that $O(1/T)$ holds for a wide range of no-swap regret algorithms.

# 6 Discussion

In this paper, we studied the convergence rate of regrets of the Hedge algorithm and its optimistic variant in two-player games. We obtained a strict separation between vanilla Hedge and optimistic Hedge, i.e., $1/\sqrt{T}$ vs. $1/T^{5/6}$. We also initiated the study on algorithms with faster convergence rates of swap regrets in general multiplayer games and obtained an algorithm with average regret $O(m^{1/2}(n \log n/T)^{3/4})$, improving over the classic result of Blum and Mansour [6].

Our work led to several interesting future directions:

- Our faster convergence result for optimistic Hedge currently only works for two-player games. Can we extend it to multiplayer games? Second, what is the optimal convergence rate for optimistic Hedge and other no-regret algorithms? even for two-player games?

- Regarding swap regrets, it is easy to generalize the result in Section 5 to any algorithm that (1) satisfies the RVU property and (2) makes only multiplicative changes on strategies each iteration. These include optimistic Hedge and optimistic multiplicative weights. However, our current analysis does not apply to general optimistic Mirror Descent or Follow the Regularized Leader. Can we still prove faster convergence of swap regrets via the reduction of [6] without requiring (2) on the regret minimization algorithm? or does there exist some natural gap between these algorithms and optimistic Hedge/multiplicative weights?

- Can we achieve similar convergence rates under partial information models? such as those considered in [24, 14, 29].

## Broader Impact

The paper has impact on research in the game theory community, and could also benefit future research on multi-agent learning.

## Acknowledgement

Xi Chen is supported by NSF IIS-1838154 and NSF CCF-1703925. Binghui Peng would thank Christos H. Papadimitriou for useful discussions.

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
