[Supplementary Material]

# Hedging in games: Faster convergence of external and swap regrets

## Abstract

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

- For our result in Appendix E on the convergence to the approximately optimal social welfare, can this fast convergence result be extended to the (exact) optimal social welfare setting (follow the definition in [27])?

- Can we achieve similar convergence rates under partial information models? such as those considered in [24, 14, 29].

## Footnotes

[1]Note that $Q^t$ used in `BM-Optimistic-Hedge` is always ergodic.

[2]The algorithm is not efficient in general. However, we can turn it into an effiecient one by considering only $n^2$ swap matrices that are equal to indentical mapping *except* for one coordinate. The regret bound will only blow up by a $\sqrt{n}$ factor.

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

 # A  Missing proof from Section 3

 **Proof of Lemma 3.2**  For each $t \in [2 : T]$, we apply Pinsker's inequality to have

$$
\frac{1}{2} \cdot \|x^t - x^{t-1}\|_1^2 \leq D_{\text{KL}}(x^{t-1}\|x^t) = \sum_{i \in [n]} x^{t-1}(i) \cdot \log\left(\frac{x^{t-1}(i)}{x^t(i)}\right)
$$

$$
= \sum_{i \in [n]} x^{t-1}(i) \cdot \log\left(\sum_{j \in [n]} \exp\left(-\eta\big(2\ell^{t-1}(j) - \ell^{t-2}(j)\big)\right) \cdot x^{t-1}(j)\right)
$$

$$
+ \sum_{i \in [n]} x^{t-1}(i) \cdot \eta\big(2\ell^{t-1}(i) - \ell^{t-2}(i)\big)
$$

$$
= \log\left(\sum_{j \in [n]} \exp\left(-\eta\big(2\ell^{t-1}(j) - \ell^{t-2}(j)\big)\right) \cdot x^{t-1}(j)\right) + \eta\langle x^{t-1}, 2\ell^{t-1} - \ell^{t-2}\rangle
$$

$$
\triangleq \Phi_t + \eta\langle x^{t-1}, 2\ell^{t-1} - \ell^{t-2}\rangle, \tag{9}
$$

 where we recall $\ell^0 = \mathbf{0}$. The third step follows from the updating rule of optimistic Hedge. Letting $L^t = \sum_{i \in [t]} \ell^i$, next we use induction to prove the following claim for each $k = 1, \ldots, T$:

$$
\sum_{t \in [k]} \Phi_t = \log\left(\sum_{j \in [n]} x^1(j) \cdot \exp\left(-\eta L^{k-1}(j) - \eta\ell^{k-1}(j)\right)\right). \tag{10}
$$

 The base case holds trivially, as $\Phi_1 = 0$. Suppose the above holds for $k$. Then for $k+1$ we have

$$
\sum_{t=1}^{k+1} \Phi_t = \sum_{t=1}^{k} \Phi_t + \Phi_{k+1}
$$

$$
= \log\left(\sum_{j \in [n]} x^1(j) \cdot \exp\left(-\eta L^{k-1}(j) - \eta\ell^{k-1}(j)\right)\right) + \log\left(\sum_{i \in [n]} \exp\left(-\eta\big(2\ell^k(i) - \ell^{k-1}(i)\big)\right) \cdot x^k(i)\right)
$$

$$
= \log\left(\left(\sum_{i \in [n]} \exp\left(-\eta\big(2\ell^k(i) - \ell^{k-1}(i)\big)\right) \cdot x^k(i)\right) \cdot \left(\sum_{j \in [n]} x^1(j) \cdot \exp\left(-\eta L^{k-1}(j) - \eta\ell^{k-1}(j)\right)\right)\right)
$$

$$
= \log\left(\sum_{i \in [n]} \exp\left(-\eta\big(2\ell^k(i) - \ell^{k-1}(i)\big)\right) \cdot x^1(i) \cdot \exp\left(-\eta L^{k-1}(i) - \eta\ell^{k-1}(i)\right)\right)
$$

$$
= \log\left(\sum_{i \in [n]} x^1(i) \cdot \exp\left(-\eta L^k(i) - \eta\ell^k(i)\right)\right),
$$

 where the third step follows from

$$
x^k(i) = \frac{x^1(i) \cdot \exp\left(-\eta L^{k-1}(i) - \eta\ell^{k-1}(i)\right)}{\sum_{j \in [n]} x^1(j) \cdot \exp\left(-\eta L^{k-1}(j) - \eta\ell^{k-1}(j)\right)}.
$$

Now we have (recall that $\Phi_1 = 0$)

$$
\begin{aligned}
\frac{1}{2\ln 2}\sum_{t\in[2:T]}\|x^t - x^{t-1}\|_1^2 &\leq \sum_{t\in[2:T]}\Big(\Phi_t + \eta\langle x^{t-1}, 2\ell^{t-1} - \ell^{t-2}\rangle\Big) \\
&= \log\left(\sum_{j\in[n]}\frac{1}{n}\cdot\exp\big(-\eta L^{T-1}(j) - \eta\ell^{T-1}(j)\big)\right) + \sum_{t\in[2:T]}\eta\langle x^{t-1}, 2\ell^{t-1} - \ell^{t-2}\rangle \\
&\leq -\min_{j\in[n]}\Big(\eta L^{T-1}(j) + \eta\ell^{T-1}(j)\Big) + \sum_{t\in[2:T]}\eta\langle x^{t-1}, 2\ell^{t-1} - \ell^{t-2}\rangle \\
&\leq -\eta\min_{j\in[n]}L^{T-1}(j) + \eta\sum_{t\in[T-1]}\langle x^t, \ell^t\rangle + \eta\sum_{t\in[T-1]}\langle x^t, \ell^t - \ell^{t-1}\rangle \\
&\leq \eta\left(\frac{2\log n}{\eta} + \eta\sum_{t\in[T-1]}\|\ell^t - \ell^{t-1}\|_\infty^2\right) + \eta\sum_{t\in[T-1]}\langle x^t, \ell^t - \ell^{t-1}\rangle \\
&\leq 2\log n + \eta^2\sum_{t\in[T-1]}\|\ell^t - \ell^{t-1}\|_\infty^2 + \eta\sum_{t\in[T-1]}\|\ell^t - \ell^{t-1}\|_\infty \\
&\leq 2\log n + (\eta + \eta^2)\sum_{t\in[T-1]}\|\ell^t - \ell^{t-1}\|_\infty.
\end{aligned}
$$

The first step follows from Eq. (9) and the second step follows from Eq. (10). The fifth step follows from Lemma 2.1. This finishes the proof of the lemma.

## B Missing proof from Section 4

### B.1 Case when the learning rate is small

We handle the case when $\eta \leq 64/(c_0\sqrt{T}) = O(1/\sqrt{T})$ with the following lemma:

**Lemma B.1.** *Suppose both players run vanilla Hedge on game $G_2 = (A, B_2)$ with learning rate $\eta = O(1/\sqrt{T})$. Then the regret of the first player is at least $\Omega(\sqrt{T})$ after $T$ rounds.*

*Proof.* The loss of player 2 is invariant to the strategy of player 1. Thus her strategy stays at $(0.4, 0.6)$. Hence, for any $t \in [T]$, the loss for player 1 is always $\ell = (-0.2, 0.2)$ and we have

$$
x^t(1) = \frac{0.4\cdot\exp(0.2\eta t)}{0.4\cdot\exp(0.2\eta t) + 0.6\cdot\exp(-0.2\eta t)} \quad\text{and}
$$

$$
x^t(2) = \frac{0.6\cdot\exp(-0.2\eta t)}{0.4\cdot\exp(0.2\eta t) + 0.6\cdot\exp(-0.2\eta t)}.
$$

One can verify that when $t \leq 1/2\eta$, we have $x^t(1) \leq 0.5 \leq x^t(2)$. Therefore, the regret is

$$
\mathrm{regret}_T^x = \sum_{t\in[T]}\langle x^t, \ell\rangle - \sum_{t\in[T]}\ell(1) \geq \sum_{t=1}^{1/2\eta}\langle x^t, \ell\rangle - \sum_{t=1}^{1/2\eta}\ell(1) \geq 0 + \frac{1}{2\eta}\cdot 0.2 = \Omega(\sqrt{T}).
$$

Thus we complete the proof. $\square$

### B.2 Case when the learning rate is large

We next work on the case when $\eta \geq 3$. Recall that we write $x_t = x^t(1)$ and $y_t = y^t(1)$.

**Lemma B.2.** *Suppose both players run vanilla Hedge on game $G_3 = (A, B_3)$ with learning rate $\eta \geq 3$ Then the regret of the first player is at least $\Omega(T)$ after $T$ rounds.*

*Proof.* Intuitively, $(A, B_3)$ is a cooperation game, and it is beneficial for both players if they choose to cooperate on one single action (by playing either $(1, 2)$ or $(2, 1)$). However, when the learning rate is too large, they actually mismatch in every iterations. Formally, we have

$$
\begin{aligned}
x_{t+1} &= \frac{x_t \cdot \exp(\eta(1 - 2y_t))}{x_t \cdot \exp(\eta(1 - 2y_t)) + (1 - x_t) \cdot \exp(\eta(2y_t - 1))} \\
&= \frac{x_t \cdot \exp(\eta(1 - 2x_t))}{x_t \cdot \exp(\eta(1 - 2x_t)) + (1 - x_t) \cdot \exp(\eta(2x_t - 1))}.
\end{aligned}
$$

The second step follows from $x_t = y_t$ for all $t$ because $A = B_3$ in the game. Motivated by this, we define a sequence $a_0, a_1, \dots$ where $a_0 = x_0 = 0.4$ and

$$
a_{t+1} = \frac{(1 - a_t) \cdot \exp(\eta(2a_t - 1))}{a_t \cdot \exp(\eta(1 - 2a_t)) + (1 - a_t) \cdot \exp(\eta(2a_t - 1))}, \quad \text{for each } t \geq 0.
$$

Then $a_t = x_t$ if $t$ is even and $a_t = 1 - x_t$ when $t$ is odd. Furthermore, by Claim B.3 below, we have $\eta \exp(-2\eta) \leq a_t \leq 0.4$ for all $t$ when $\eta \geq 3$. Hence, we have

$$
\text{regret}_T^x \geq \sum_{t \in [T]} \langle x^t, \ell_x^t \rangle = \sum_{t \in [T]} (2x_t - 1)^2 = \sum_{t \in [T]} (2a_t - 1)^2 \geq \Omega(T).
$$

This finishes the proof of the lemma. □

**Claim B.3.** *When $\eta \geq 3$, we have $\eta \exp(-2\eta) \leq a_t \leq 0.4$ for all $t \geq 0$.*

*Proof.* We prove by induction on $t$. The base case holds trivially for $t = 0$. Suppose the inequality holds up to $t$. Then for $t + 1$, we have

$$
\frac{a_{t+1}}{1 - a_{t+1}} = \frac{1 - a_t}{a_t} \cdot \exp\left(\eta(4a_t - 2)\right) \triangleq f(a_t).
$$

By simple calculation, we know that $f(a_t)$ takes maximium at $\eta \exp(-2\eta)$ or $0.4$. Thus,

$$
\frac{a_{t+1}}{1 - a_{t+1}} \leq \max\left\{f(0.4), f(\eta \exp(-2\eta))\right\} \leq \frac{2}{3},
$$

which implies that $a_{t+1} \leq 0.4$. The second step above follows from

$$
f(0.4) = \frac{3}{2} \cdot \exp(-0.4\eta) \leq \frac{2}{3},
$$

using $\eta \geq 3$ and

$$
f\left(\eta \exp(-2\eta)\right) \leq \frac{1}{\eta} \exp(2\eta) \cdot \exp\left(4\eta^2 \exp(-2\eta) - 2\eta\right) = \frac{1}{\eta} \cdot \exp\left(4\eta^2 \exp(-2\eta)\right) \leq \frac{2}{3}.
$$

Moreover, $f(a_t)$ takes minimum at the smaller solution $a$ of $4\eta a(1 - a) = 1$. Thus,

$$
\frac{a_{t+1}}{1 - a_{t+1}} \geq \frac{1 - a}{a} \cdot \exp\left(\eta(4a - 2)\right) \geq \frac{4}{3} \cdot \eta \exp(-2\eta),
$$

where the second step used $\exp(\eta(4a - 2)) \geq \exp(-2\eta)$, $a \leq 1/2\eta$ and $a \leq 1/3$. This shows that $a_{t+1} \geq \eta \exp(-2\eta)$ using $\eta \geq 3$, and finishes the induction. □

### B.3 Proof of Lemma 4.3

Note that the Matching Pennies game $G_1 = (A, B_1)$ is zero-sum. It is known (see [5]) that the KL divergence of vanilla Hedge in zero-sum games is strictly increasing. We give a careful analysis on its increment each round when playing $G_1$. (Recall that $x^\star = y^\star = (0.5, 0.5)$.)

**Lemma B.4.** *Suppose both players run vanilla Hedge with $\eta \leq 3$ on $G_1$. Then for each $t \geq 0$,*

$$
\begin{aligned}
&D_{KL}(x^\star \| x^{t+1}) + D_{KL}(y^\star \| y^{t+1}) - \left(D_{KL}(x^\star \| x^t) + D_{KL}(y^\star \| y^t)\right) \\
&\qquad \geq e^{-7}\eta^2 x_t(1 - x_t)(2y_t - 1)^2 + e^{-7}\eta^2 y_t(1 - y_t)(2x_t - 1)^2.
\end{aligned}
$$

*Proof.* Focusing on the first player, we have

$$D_{\text{KL}}(x^\star \| x^{t+1}) - D_{\text{KL}}(x^\star \| x^t)$$

$$= \sum_{i \in [2]} x^\star(i) \cdot \log\left(\frac{x^\star(i)}{x^{t+1}(i)}\right) - \sum_{i \in [2]} x^\star(i) \cdot \log\left(\frac{x^\star(i)}{x^t(i)}\right)$$

$$= \sum_{i \in [2]} x^\star(i) \cdot \log\left(\frac{x^t(i)}{x^{t+1}(i)}\right)$$

$$= \sum_{i \in [2]} x^\star(i) \cdot \eta \ell^t(i) + \sum_{i \in [2]} x^\star(i) \cdot \log\left(\sum_{j \in [2]} x^t(j) \cdot \exp(-\eta \ell^t(j))\right)$$

$$= \log\left(\sum_{j \in [2]} x^t(j) \cdot \exp(-\eta \ell^t(j))\right)$$

$$= \log\left(x_t \cdot \exp(-\eta(2y_t - 1)) + (1 - x_t) \cdot \exp(-\eta(1 - 2y_t))\right)$$

$$\geq x_t \cdot (-\eta(2y_t - 1)) + (1 - x_t) \cdot (-\eta(1 - 2y_t)) + \frac{1}{2e^6} x_t(1 - x_t)\left(e^{-\eta(2y_t-1)} - e^{-\eta(1-2y_t)}\right)^2$$

$$\geq \eta(2y_t - 1)(1 - 2x_t) + e^{-7}\eta^2 x_t(1 - x_t)(2y_t - 1)^2. \tag{11}$$

The third step follows from the updating rule of vanilla Hedge. The fourth step uses $x^\star(1) = x^\star(2) = 0.5$ and $\ell^t(1) + \ell^t(2) = (2y_t - 1) + (1 - 2y_t) = 0$. The sixth step uses the fact that $f(x) = -\log x$ is $e^{-6}$-strongly convex on $(0, e^3)$. Similarly, we can prove

$$D_{\text{KL}}(y^\star \| y^{t+1}) - D_{\text{KL}}(y^\star \| y^t) \geq \eta(2x_t - 1)(2y_t - 1) + e^{-7}\eta^2 y_t(1 - y_t)(2x_t - 1)^2. \tag{12}$$

The lemma follows by combining (11) and (12). $\qquad\square$

We are now ready to prove Lemma 4.3.

*Proof of Lemma 4.3.* We first prove that within $O(1/\eta^2)$ steps, the KL divergence $D_{\text{KL}}(x^\star \| x^t) + D_{\text{KL}}(y^\star \| y^t)$ becomes at least 20. The proof follows directly from Lemma B.4, as for any $t$ with $D_{\text{KL}}(x^\star \| x^t) + D_{\text{KL}}(y^\star \| y^t) \leq 20$, we have

$$D_{\text{KL}}(x^\star \| x^{t+1}) + D_{\text{KL}}(y^\star \| y^{t+1}) - \left(D_{\text{KL}}(x^\star \| x^t) + D_{\text{KL}}(y^\star \| y^t)\right)$$
$$\geq e^{-7}\eta^2 x_t(1 - x_t)(2y_t - 1)^2 + e^{-7}\eta^2 y_t(1 - y_t)(2x_t - 1)^2 \geq \Omega(\eta^2). \tag{13}$$

The second step follows from the fact that both $x_t$ and $y_t$ are bounded away from 0 and 1 given the divergence at $t$ is at most 20; it also used $\max\{|2x_t - 1|, |2y_t - 1|\} \geq 0.2$ given that the divergence is strictly increasing.

Let $T_0 = O(1/\eta^2)$ be the first time when the divergence becomes at least 20. If $T/2 \leq T_0$, it follows from (13) that the divergence at $T$ is $\Omega(T\eta^2) = \Omega(\sqrt{T}\eta)$ using the assumption that $\eta \geq 16/\sqrt{T}$. So we focus on the case $T_0 \leq T/2$ and thus, $T = T_0 + L$ with $L \geq T/2$. We prove

**Claim B.5.** *At round $t = T_0 + \tau^2$, the KL divergence has $D_{KL}(x^\star \| x^t) + D_{KL}(y^\star \| y^t) \geq 10^{-10}\tau\eta$.*

Setting $\tau = \sqrt{T/2}$ so that $T_0 + \tau^2 \leq T$, we have

$$D_{\text{KL}}(x^\star \| x^T) + D_{\text{KL}}(y^\star \| y^T) \geq \Omega(\sqrt{T}\eta),$$

and this finishes the proof of the lemma. $\qquad\square$

*Proof of Claim B.5.* We proceed to use induction on $\tau$. The cases with $\tau \leq 16/\eta$ holds trivially as the KL divergence at $T_0$ is already at least 20. For the induction step, suppose the claim holds up to $k$ for some $k \geq 64/\eta$ at time $t_0 = T_0 + k^2$. We show that at time $T_0 + (k + 1)^2$ the KL divergence is at least $10^{-10}(k + 1)\eta$. Without loss of generality, we assume that $x_{t_0}, y_{t_0} \geq 0.5$; the other three cases can be handled similarly. In this region, $x_t$ with $t = t_0 + 1, \ldots$ will keep decreasing and $y_t$ will keep increasing, until the moment when $x_t$ drops below 0.5.

Let $t_2$ denote the first round $t_2 > t_0$ such that $x_t \leq 0.5$. We first show that it will take no more than $k/2$ rounds for $x_t$ to drop below 0.5: $t_2 - t_0 \leq k/2$. To this end, we use $t_1$ to denote the first round $t_1 \geq t_0$ such that $y_t \geq 3/4$ and note that $t_1 \leq t_2$ (since otherwise at $t = t_2 - 1$, we have $1/2 \leq y_t \leq 3/4$ and $1/2 \leq x_t \leq e^6$ in order for $x_t$ to go below $1/2$ with $\eta \leq 3$ in the next round; this contradicts with the fact that the KL divergence is at least 20 after $T_0$).

We break the proof of $t_2 - t_0 \leq k/2$ into two phases: $t_1 - t_0 \leq k/4$ and $t_2 - t_1 \leq k/4$.

**Phase 1.** First we prove that it takes no more than $k/4$ steps for $y_t$ to get larger than $3/4$. To this end, we notice that for all $t \in [t_0 : t_1 - 1]$, we have $y_t \leq 3/4$ and thus, $x_t \geq 3/4$ since the KL divergence is at least 20. During all these rounds the loss vector $\ell_y^t$ of the second player satisfies $\ell_y^t(1) \leq -3/4 + 1/4 \leq -0.5$ and $\ell_y^t(2) \geq 0.5$. Thus we have (using $0.5 \leq y_{t_0} \leq y_{t_1-1} \leq 3/4$)

$$3 \geq \frac{y_{t_1-1}}{1 - y_{t_1-1}} \geq \exp\left(\eta(t_1 - t_0 - 1)\right) \cdot \frac{y_{t_0}}{1 - y_{t_0}} \geq \exp\left(\eta(t_1 - t_0 - 1)\right).$$

Thus $t_1 - t_0 \leq (2/\eta) + 1 \leq k/4$ using $k \geq 64/\eta$ and $\eta \leq 3$.

**Phase 2.** Next we prove that, starting from $t_1$, it takes less than $k/4$ steps for $x_t$ to drop below 0.5. Note that for each $t \in [t_1 : t_2 - 1]$, the loss vector $\ell_x^t$ of the first player satisfies $\ell_x^t(1) \geq 0.5$ and $\ell_x^t(2) \leq -0.5$. Moreover, we assume without loss of generality that $1 - x_{t_1} \geq \exp(-(k+1)\eta/20)$; otherwise the KL divergence at $t_1$ is already bigger than $10^{-10}(k+1)\eta$ and we are done. Therefore,

$$1 \leq \frac{x_{t_2-1}}{1 - x_{t_2-1}} \leq \exp\left(-\eta(t_2 - t_1 - 1)\right) \cdot \frac{x_{t_1}}{1 - x_{t_1}} \leq \exp\left(\eta(-(t_2 - t_1 - 1) + (k+1)/20)\right)$$

Thus $t_2 - t_1 \geq 1 + (k+1)/20 \leq k/4$ using $k \geq 64/\eta \geq 64/3$.

Now we are at time $t_2$ and we examine the next $R = 3/\eta \leq k/2$ rounds $[t_2 : t_2 + R]$; these are the rounds where we will gain a lot in the KL divergence. Given that $x_{t_2}$ just dropped below $1/2$, we have $x_{t_2} \geq 0.5 \cdot \exp(-2\eta)$ and thus, for every $t \in [t_2 : t_2 + R]$,

$$x_t \geq x_{t_2} \cdot \exp(-2\eta \cdot R) \geq 0.5 \cdot e^{-12}.$$

Consequently, we have

$$\left(D_{\mathrm{KL}}(x^\star \| x^{t_2+R}) + D_{\mathrm{KL}}(y^\star \| y^{t_2+R})\right) - \left(D_{\mathrm{KL}}(x^\star \| x^{t_2}) + D_{\mathrm{KL}}(y^\star \| y^{t_2})\right)$$

$$\geq \sum_{t=t_2}^{t_2+R-1} e^{-7}\eta^2 x_t(1-x_t)(2y_t-1)^2 + e^{-7}\eta^2 y_t(1-y_t)(2x_t-1)^2$$

$$\geq \sum_{t=t_2}^{t_2+R-1} e^{-7}\eta^2 x_t(1-x_t)(2y_t-1)^2 \geq \frac{3}{\eta} \cdot e^{-7}\eta^2 \cdot \frac{1}{4}e^{-12} \cdot \frac{1}{4} \geq 10^{-10}\eta.$$

So we conclude that after at most $k/4 + k/4 + k/2 = k$ steps, the KL divergence increase at least $10^{-10}\eta$. Thus at time $T_0 + k^2 + k \leq T_0 + (k+1)^2$, the KL divergence is at least $10^{-10}k\eta + 10^{-10}\eta = 10^{-10}(k+1)\eta$. This finishes the induction and the proof of the claim. $\qquad\square$

# C   Missing proof from Section 5

## C.1   Proof of Lemma 5.2

Fix any swap function $\phi : [n] \to [n]$. By Lemma 2.1, every $\mathtt{ALG}_j$ achieves low regret. Thus,

$$\sum_{t \in [T]} \langle q_j^t, x^t(j)\ell^t \rangle \leq \sum_{t \in [T]} x^t(j) \cdot \ell^t(\phi(j)) + \frac{2\log n}{\eta} + \eta \sum_{t \in [T]} \|x^t(j)\ell^t - x^{t-1}(j)\ell^{t-1}\|_\infty^2, \quad (14)$$

where we used $x^t = Q^t x^t$, set $\ell^0 = \mathbf{0}$ and $x^0 = \mathbf{1}/n = x^1$. Consequently, we have

$$\sum_{t\in[T]} \langle x^t, \ell^t \rangle = \sum_{t\in[T]} \langle x^t Q^t, \ell^t \rangle = \sum_{t\in[T]} \sum_{j\in[n]} \langle x^t(j) q_j^t, \ell^t \rangle = \sum_{j\in[n]} \sum_{t\in[T]} \langle q_j^t, x^t(j)\ell^t \rangle$$

$$\le \sum_{j\in[n]} \left( \sum_{t\in[T]} x^t(j) \cdot \ell^t(\phi(j)) + \frac{2\log n}{\eta} + \eta \sum_{t\in[T]} \|x^t(j)\ell^t - x^{t-1}(j)\ell^{t-1}\|_\infty^2 \right)$$

$$= \sum_{t\in[T]} \sum_{j\in[n]} x^t(j) \cdot \ell^t(\phi(j)) + \frac{2n\log n}{\eta} + \eta \sum_{t\in[T]} \sum_{j\in[n]} \|x^t(j)\ell^t - x^{t-1}(j)\ell^{t-1}\|_\infty^2$$

where the first inequality follows from (14). Furthermore, we have (using $\|\ell^t\|_\infty \le 1$ and $\|x^t\|_1 = 1$)

$$\sum_{j\in[n]} \|x^t(j)\ell^t - x^{t-1}(j)\ell^{t-1}\|_\infty^2 \le \sum_{j\in[n]} \left( \|x^t(j)\ell^t - x^{t-1}(j)\ell^t\|_\infty + \|x^{t-1}(j)\ell^t - x^{t-1}(j)\ell^{t-1}\|_\infty \right)^2$$

$$\le 2 \sum_{j\in[n]} \|x^t(j)\ell^t - x^{t-1}(j)\ell^t\|_\infty^2 + 2 \sum_{j\in[n]} \|x^{t-1}(j)\ell^t - x^{t-1}(j)\ell^{t-1}\|_\infty^2$$

$$= 2 \sum_{j\in[n]} \left( x^t(j) - x^{t-1}(j) \right)^2 \|\ell^t\|_\infty^2 + 2 \sum_{j\in[n]} (x^{t-1}(j))^2 \|\ell^t - \ell^{t-1}\|_\infty^2$$

$$= 2 \left( \|x^t - x^{t-1}\|_2^2 \cdot \|\ell^t\|_\infty^2 + \|x^{t-1}\|_2^2 \cdot \|\ell^t - \ell^{t-1}\|_\infty^2 \right)$$

$$\le 2 \left( \|x^t - x^{t-1}\|_1^2 + \|\ell^t - \ell^{t-1}\|_\infty^2 \right)$$

We can combine all these inequalities (and note that $x^0 = x^1$) to finish the proof of the lemma.

## C.2 Proof of Lemma 5.3

We start the proof of Lemma 5.3 with the following definition.

**Definition C.1.** *Given Markov chains $Q, Q' \in \mathbb{R}^{n\times n}$, we say $Q'$ is $(\eta_1, \ldots, \eta_n)$-approximate to $Q$ if $(1 - \eta_i)q'_{i,j} \le q_{i,j} \le (1 + \eta_i)q'_{i,j}$ for every $i, j \in [n]$, where we write $Q = (q_{i,j})$ and $Q' = (q'_{i,j})$.*

We are ready to state our perturbation analysis on ergodic[1] Markov chains.

**Lemma C.2.** *Given two ergodic Markov chains $Q$ and $Q'$, where $Q'$ is $(\eta_1, \ldots, \eta_n)$-approximate to $Q$, the stationary distribution $p, p'$ of $Q$ and $Q'$, respectively, satisfy $\|p - p'\|_1 \le 8 \sum_{i=1}^{n} \eta_i$.*

The proof of Lemma C.2 relies on the classical Markov chain tree theorem (see [1]). To state it we need the following definition.

**Definition C.3.** *Suppose $Q$ is an ergodic Markov chain and $G = (V, E)$ with $V = [n]$ is the weighted directed graph associated with $Q$. We say a subgraph $T$ of $G$ is a* directed tree rooted at $i \in [n]$ *if (1) $T$ does not contain any cycles and (2) Node $i$ has no outgoing edges, while every other node $j \in [n]$ has exactly one outgoing edge. For each node $i \in [n]$, we write $\mathcal{T}_i$ to denote the set of all directed trees rooted at node $i$. We further define*

$$\Sigma_i = \sum_{T\in\mathcal{T}_i} \prod_{(a,b)\in T} q_{a,b} \quad \text{and} \quad \Sigma = \sum_{i\in[n]} \Sigma_i,$$

*i.e., the weight of $T$ is the product of its edge weights and $\Sigma_i$ is the sum of weights of trees in $\mathcal{T}_i$.*

We can now formally state the Markov chain tree theorem.

**Theorem C.4** (Markov chain tree theorem; see [1]). *Suppose $Q$ is an erogidc Markov chain and $p$ is its stationary distribution. Then we have $p_i = \Sigma_i / \Sigma$ for every $i \in [n]$.*

We now use the Markov chain tree theorem to prove Lemma C.2.

*Proof of Lemma C.2.* Note that the lemma is trivial when $\sum_{i=1}^{n} \eta_i > 1/4$ so we assume without loss of generality that $\sum_{i=1}^{n} \eta_i \leq 1/4$. For any $i \in [n]$, we have

$$\Sigma_i = \sum_{T \in \mathcal{T}_i} \prod_{(a,b) \in T} q_{a,b} \leq \sum_{T \in \mathcal{T}_i} \prod_{(a,b) \in T} (1 + \eta_a) \widetilde{q}_{a,b}$$

$$\leq \prod_{j \in [n]} (1 + \eta_j) \sum_{T \in \mathcal{T}_i} \prod_{(a,b) \in T} q'_{a,b} = \prod_{j \in [n]} (1 + \eta_j) \cdot \Sigma'_i \leq \left( 1 + 2 \sum_{j \in [n]} \eta_j \right) \Sigma'_i. \quad (15)$$

The third step holds because for any tree $T \in \mathcal{T}_i$, each node, other than node $i$, appears exactly once as $a$ when calculating the weight of $T$. The last step follows from the fact that when $\sum_{i=1}^{n} \eta_i \leq 1/4$,

$$\prod_{j \in [n]} (1 + \eta_j) \leq \prod_{j \in [n]} e^{\eta_j} = e^{\sum_{j \in [n]} \eta_j} \leq 1 + 2 \sum_{j \in [n]} \eta_j.$$

Similarly, we have

$$\Sigma_i \geq \sum_{T \in \mathcal{T}_i} \prod_{(a,b) \in T} (1 - \eta_a) \widetilde{q}_{a,b} \geq \prod_{j \in [n]} (1 - \eta_j) \cdot \Sigma'_i \geq \left( 1 - 2 \sum_{j \in [n]} \eta_j \right) \Sigma'_i. \quad (16)$$

The last inequality holds since, for $\sum_{j=1}^{n} \eta_j \leq 1/2$, we have

$$\prod_{j \in [n]} (1 - \eta_j) \geq \prod_{j \in [n]} e^{-2\eta_j} = \exp\left( -2 \sum_{j \in [n]} \eta_j \right) \geq 1 - 2 \sum_{j \in [n]} \eta_j.$$

Since $\Sigma = \sum_i \Sigma_i$, we have $(1 - 2 \sum_i \eta_i) \widetilde{\Sigma} \leq \Sigma \leq (1 + 2 \sum_i \eta_i) \widetilde{\Sigma}$. Applying Theorem C.4,

$$\|p - p'\|_1 = \sum_{i \in [n]} |p_i - p'_i| = \sum_{i \in [n]} \left| \Sigma_i / \Sigma - \Sigma_i' / \Sigma' \right| \leq \sum_{i \in [n]} \left| \Sigma_i / \Sigma - \Sigma_i / \Sigma' \right| + \sum_{i \in [n]} \left| \Sigma_i / \Sigma' - \Sigma_i' / \Sigma' \right|$$

$$\leq \sum_{i \in [n]} \frac{2 \sum_{i=1}^{n} \eta_i}{1 - 2 \sum_{i=1}^{n} \eta_i} \left| \Sigma_i / \Sigma \right| + \sum_{i \in [n]} 2 \sum_{j \in [n]} \eta_j \cdot \left| \Sigma_i' / \Sigma' \right| \leq 6 \sum_{i \in [n]} \eta_i.$$

This finishes the proof of the lemma. $\qquad \square$

Finally we prove Lemma 5.3:

*Proof of Lemma 5.3.* We start with the following claim, which states that entries of $Q^t$ and $Q^{t-1}$ only differs by a small multiplicative factor.

**Claim C.5.** *Suppose that the learning rate $\eta \leq 1/6$ and let $x^0 = \mathbf{1}/n = x^1$. Then for any $t \geq 2$, $Q^t$ is a $(\eta_1, \ldots, \eta_n)$-approximate to $Q^{t-1}$, where $\eta_j = 2\eta x^{t-2}(j) + 4\eta x^{t-1}(j)$ for each $j \in [n]$.*

Combing Claim C.5 and Lemma C.2, we have

$$\|x^t - x^{t-1}\|_1 \leq 8 \sum_{j \in [n]} \eta_j = 8 \sum_{j \in [n]} \left( 2x^{t-2}(j) + 4x^{t-1}(j) \right) \eta = 48\eta.$$

This finishes the proof of Lemma 5.3. $\qquad \square$

*Proof of Claim C.5.* Let $x^0 = \mathbf{1}/n = x^1$. By the updating rule of optimisitic Hedge, we have for any $t \geq 2$, $i, j \in [n]$ that

$$q_j^t(i) = \frac{\exp(-\eta(2x^{t-1}(j)\ell^{t-1}(i) - x^{t-2}(j)\ell^{t-2}(i))) \cdot q_j^{t-1}(i)}{\sum_{k \in [n]} \exp(-\eta(2x^{t-1}(j)\ell^{t-1}(k) - x^{t-2}(j)\ell^{t-2}(k))) \cdot q_j^{t-1}(k)}$$

$$\leq \frac{\exp(\eta x^{t-2}(j)) \cdot q_j^{t-1}(i)}{\sum_{k \in [n]} \exp(-2\eta x^{t-1}(j)) \cdot q_j^{t-1}(k)}$$

$$= \exp\left( \eta x^{t-2}(j) + 2\eta x^{t-1}(j) \right) \cdot q_j^{t-1}(i)$$

$$\leq (1 + 2\eta x^{t-2}(j) + 4\eta x^{t-1}(j)) \cdot q_j^{t-1}(i).$$

The second step follows from $\ell^t \in [0,1]^n$ and the last step follows from $\exp(a) \le 1 + 2a$ for $a \le 1/2$.

The other side holds similarly:

$$q_j^t(i) = \frac{\exp(-\eta(2x^{t-1}(j)\ell^{t-1}(i) - x^{t-2}(j)\ell^{t-2}(i))) \cdot q_j^{t-1}(i)}{\sum_{k \in [n]} \exp(-\eta(2x^{t-1}(j)\ell^{t-1}(k) - x^{t-2}(j)\ell^{t-2}(k))) \cdot q_j^{t-1}(k)}$$

$$\ge \frac{\exp(-2\eta x^{t-1}(j)) \cdot q_j^{t-1}(i)}{\sum_{k \in [n]} \exp(\eta x^{t-2}(j)) \cdot q_j^{t-1}(k)}$$

$$= \exp\left(-\eta x^{t-2}(j) - 2\eta x^{t-1}(j)\right) \cdot q_j^{t-1}(i)$$

$$\ge (1 - \eta x^{t-2}(j) - 2\eta x^{t-1}(j)) \cdot q_j^{t-1}(i).$$

Thus completing the proof. $\qquad\qquad\qquad\qquad\qquad\qquad\qquad\qquad\qquad\qquad\qquad\quad \square$

## C.3  Proof of Corollary 5.4

The algorithm works as follow. We set

$$\eta = \frac{(n \log n)^{1/4}}{m^{1/2} T^{1/4}}$$

and $B_r = 1$ at initialization, for any player $i \in [m]$ and $\tau = 1, \ldots, T$

1. Play $x_i^t$ according to BM-Optimistic-Hedge, and receive $\ell_i^t$.

2. If $\sum_{t=2}^{\tau} \|\ell_i^t - \ell_i^{t-1}\|_\infty^2 + \sum_{t=2}^{\tau} \|x_i^t - x_i^{t-1}\|_1^2 \ge B_r$.

   (a) Update $B_{r+1} = 2B_r$, $r \leftarrow r+1$, $\eta_r = \min\left\{\sqrt{\frac{n \log n}{B_r}}, \eta\right\}$.

   (b) Start a new run of BM-Optimistic-Hedge with learning rate $\eta_r$.

For any round $r$, we use $T_r$ to denote its final iteration and

$$I_r = \sum_{t=T_{r-1}+1}^{T_r} \|x_i^t - x_i^{t-1}\|_1^2 + \sum_{t=T_{r-1}+1}^{T_r} \|\ell_i^t - \ell_i^{t-1}\|_\infty^2.$$

Then we have

$$\text{swap-regret}_{T_{r-1}+1:T_r} \le \frac{2n \log n}{\eta_r} + 2\eta_r \left( \sum_{t=T_{r-1}+1}^{T_r} \|x_i^t - x_i^{t-1}\|_1^2 + \sum_{t=T_{r-1}+1}^{T_r} \|\ell_i^t - \ell_i^{t-1}\|_\infty^2 \right)$$

$$\le 2(n \log n)^{3/4} \cdot T^{1/4} m^{1/2} + 2\sqrt{n \log n B_r} + 2\eta_r \cdot I_r$$

$$\le 2(n \log n)^{3/4} \cdot T^{1/4} m^{1/2} + 2\sqrt{n \log n B_r} + 2\sqrt{2n \log n I_r}$$

$$\le 2(n \log n)^{3/4} \cdot T^{1/4} m^{1/2} + 4\sqrt{2n \log n I_r}$$

$$\le 2(n \log n)^{3/4} \cdot T^{1/4} m^{1/2} + 4\sqrt{2n \log n} \cdot \sqrt{\left( \sum_{t=2}^{T} \|x_i^t - x_i^{t-1}\|_1^2 + \sum_{t=2}^{T} \|\ell_i^t - \ell_i^{t-1}\|_\infty^2 \right)}$$

The first step follows from Lemma 5.2, the second step follows from the definition of $I_r$ and the fact

$$\frac{1}{\eta_r} \le \frac{1}{\eta} + \sqrt{\frac{B_r}{n \log n}} = \frac{m^{1/2} T^{1/4}}{(n \log n)^{1/4}} + \sqrt{\frac{B_r}{n \log n}}$$

The third step follows from $\eta_r \le \sqrt{\frac{n \log n}{B_r}} \le \sqrt{\frac{n \log n}{I_r/2}}$, and the last step comes from $\sqrt{B_r} \le \sqrt{2I_r}$.

Since the number of round is at most $O(\log T)$, we have

$$\text{swap-regret}_T \le \log T \left( 2(n \log n)^{3/4} T^{1/4} m^{1/2} + 4\sqrt{2n \log n} \cdot \sqrt{2 \left( \sum_{t=1}^{T} \|x_i^t - x_i^{t-1}\|_1^2 + \sum_{t=1}^{T} \|\ell_i^t - \ell_i^{t-1}\|_\infty^2 \right)} \right)$$

If all players adopt the algorithm, then we know their learning rate is no greater than $\eta = \frac{(n\log n)^{1/4}}{m^{1/2}T^{1/4}}$, thus we know $\|x_i^t - x_i^{t-1}\|_1 \le O(\eta) = O\left(\frac{(n\log n)^{1/4}}{m^{1/2}T^{1/4}}\right)$ (see Lemma 5.3) and $\|\ell_i^t - \ell_i^{t-1}\|_\infty \le \sum_{j\neq i}\|x_j^t - x_j^{t-1}\|_1 \le m \cdot O(\eta) = O\left(\frac{m^{1/2}(n\log n)^{1/4}}{T^{1/4}}\right)$. Thus the swap regret is at most

$$O\left((n\log n)^{3/4}m^{1/2}T^{1/4}\log T\right).$$

If the player is facing an adversary, then $\|x_i^t - x_i^{t-1}\|_1 \le 2$ and $\|\ell_i^t - \ell_i^{t-1}\|_\infty \le 1$, thus we conclude its regret is at most

$$O\left(\sqrt{n\log nT}\log T + (n\log n)^{3/4}m^{1/2}T^{1/4}\log T\right).$$

# D  Another no swap regret algorithm

We prove the optimistic variant of a folklore algorithm, originally appeared in [7], could also achieve fast convergence of swap regret. Our perturbation analysis again plays a key role in the regret analysis.

Define $\Phi$ to be all swap functions that map $[n]$ to $[n]$. We have $|\Phi| = n^n$. For any $\phi \in \Phi$, define the swap matrice $S^\phi$ as: $S_{i,j}^\phi = 1$ if $\phi(i) = j$ and $S_{i,j}^\phi = 0$ otherwise. It is easy to see that $S^\phi$ contains exactly one 1 each row.

[7] treats each swap matrice $S^\phi$ as an expert, and run Hedge algorithm on all $n^n$ swap matrices. At time $t$, the output strategy $p^t$ is determined by these experts via solving a fix point problem[2]. The optimisitic variant of [7] is shown in Algorithm 1. We first analysis the regret,

---
**Algorithm 1**

---
1: **for** $t = 1, 2, \ldots,$ **do**
2:    Play $p^t$ and receive the loss vector $l^t$.
3:    Update

$$q^{t+1}(\phi) = \frac{x^t(\phi)\exp(-\eta(2x^t S^\phi \ell^t - x^{t-1}S^\phi \ell^{t-1}))}{\sum_{\phi\in\Phi}x^t(\phi)\exp(-\eta(2x^t S^\phi \ell^t - x^{t-1}S^\phi \ell^{t-1}))} \quad \forall\phi\in\Phi$$

4:    Compute $x^{t+1} = x^{t+1}Q^{(t+1)}$, where

$$Q^{(t+1)} = \sum_{\phi\in\Phi}q^{t+1}(\phi)S^\phi.$$

5: **end for**

---

**Lemma D.1.** *Algorithm 1 achieves regret*

$$\text{swap-regret}_T \le \frac{n\log n}{\eta} + 2\eta\sum_{t=2}^T \|x^t - x^{t-1}\|_1^2 + 2\eta\sum_{t=2}^T \|\ell^t - \ell^{t-1}\|_\infty^2.$$

498  *Proof.* According to the updating rule, for any $\phi \in \Phi$, we have

$$
\begin{aligned}
\text{swap-regret}_T &= \sum_{t=2}^{T} \langle x^t, \ell^t \rangle - \max_{\phi \in \Phi} \sum_{t=2}^{T} x^t S^\phi \ell^t \\
&= \sum_{t=2}^{T} \langle x^t Q^{(t)}, \ell^t \rangle - \max_{\phi \in \Phi} \sum_{t=2}^{T} x^t S^\phi \ell^t \\
&= \sum_{t=2}^{T} \sum_{\phi \in \Phi} x^t (q^t(\phi) S^\phi) \ell^t - \max_{\phi \in \Phi} \sum_{t=2}^{T} x^t S^\phi \ell^t \\
&= \sum_{t=2}^{T} \sum_{\phi \in \Phi} q^t(\phi) \cdot x^t S^\phi \ell^t - \max_{\phi \in \Phi} \sum_{t=2}^{T} x^t S^\phi \ell^t \\
&\leq \frac{n \log n}{\eta} + \eta \sum_{t=2}^{T} \max_{\phi \in \Phi} \left| x^t S^\phi \ell^{t-1} - x^{t-1} S^\phi \ell^{t-1} \right\|^2 \\
&\leq \frac{\log n}{\eta} + 2\eta \sum_{t=2}^{T} \|x^t - x^{t-1}\|_1^2 + 2\eta \sum_{t=2}^{T} \|\ell^t - \ell^{t-1}\|_\infty^2.
\end{aligned}
$$

499  The fifth step follows the regret bound of optimistic Hedge and the last step follows from the fact that
500  for any $\phi \in \Phi$,

$$
\begin{aligned}
\left| x^t S^\phi \ell^t - x^t S^\phi \ell^t \right|^2 &= \left| x^t S^\phi \ell^t - x^{t-1} S^\phi \ell^t + x^{t-1} A_\phi \ell^t - x^{t-1} S^\phi \ell^{t-1} \right|^2 \\
&\leq 2 \left| x^t S^\phi \ell^t - x^{t-1} S^\phi \ell^t \right|^2 + 2 |x^{t-1} S^\phi \ell^t - x^{t-1} S^\phi \ell^{t-1}|^2 \\
&= 2 \langle x^t - x^{t-1}, S^\phi \ell^t \rangle + 2 \langle x^{t-1} S^\phi, \ell^t - l^{t-1} \rangle \\
&\leq 2 \|x^t - x^{t-1}\|_1^2 \|S^\phi \ell^t\|_\infty^2 + 2 \|x^{t-1} S^\phi\|_1 \|\ell^t - \ell^{t-1}\|_\infty^2 \\
&\leq 2 \|x^t - x^{t-1}\|_1^2 + 2 \|\ell^t - \ell^{t-1}\|_\infty^2.
\end{aligned}
$$

501  Thus completing the proof. $\qquad\square$

502  It remains to show that the environment is stable. Again, since $x^t$ is the stationary distribution of
503  $Q^{(t)}$, we only need some perturbation analysis on $Q^{(t)}$. In particular, we have

504  **Lemma D.2.** *For any $t$, $Q^{(t)}$ is $(6\eta, \ldots, 6\eta)$ approximate to $Q^{(t+1)}$.*

505  *Proof.* For any $\phi$, we have

$$
\begin{aligned}
q^{t+1}(\phi) &= \frac{q^t(\phi) \exp(-\eta(2x^t A_\phi \ell^t - x^{t-1} A_\phi \ell^{t-1}))}{\sum_{\phi \in \Phi} q^t(\phi) \exp(-\eta(2x^t A_\phi \ell^t - x^{t-1} A_\phi \ell^{t-1}))} \\
&\leq \frac{q^t(\phi) \exp(\eta)}{\sum_{\phi \in \Phi} q^t(\phi) \exp(-2\eta)} \\
&\leq (1 + 6\eta) q^t(\phi)
\end{aligned}
$$

506  Similarly, we have

$$
\begin{aligned}
q^{t+1}(\phi) &= \frac{q^t(\phi) \exp(-\eta(2x^t A_\phi \ell^t - x^{t-1} A_\phi \ell^{t-1}))}{\sum_{\phi \in \Phi} q^t(\phi) \exp(-\eta(2x^t A_\phi \ell^t - x^{t-1} A_\phi \ell^{t-1}))} \\
&\geq \frac{q^t(\phi) \exp(-2\eta)}{\sum_{\phi \in \Phi} q^t(\phi) \exp(\eta)} \\
&\geq (1 - 6\eta) q^t(\phi)
\end{aligned}
$$

507  Thus, for any $i, j \in [n]$, we have

$$
Q_{i,j}^{(t+1)} = \sum_{\phi \in \Phi} q^{t+1}(\phi) S_{i,j}^\phi \leq (1 + 6\eta) \sum_{\phi \in \Phi} q^t(\phi) S_{i,j}^\phi = (1 + 6\eta) Q_{i,j}^{(t)}
$$

508    and

$$Q_{i,j}^{(t+1)} = \sum_{\phi \in \Phi} q^{t+1}(\phi) S_{i,j}^{\phi} \geq (1 - 6\eta) \sum_{\phi \in \Phi} q^t(\phi) S_{i,j}^{\phi} \geq (1 - 6\eta) Q_{i,j}^{(t)}$$

509    Thus we conclude $Q^{(t)}$ is $(6\eta, \ldots, 6\eta)$ approximate to $Q^{(t+1)}$.    □

510    Combining the above results, we have

511    **Theorem D.3.** *Suppose every player uses Algorithm 1 and choose $\eta = O\left((\frac{\log n}{nm^2 T})^{1/4}\right)$, then each*

512    *individual's swap regret is at most $O\left(m^{1/2} n^{5/4} (\log n)^{3/4} T^{1/4}\right)$.*

513    *Proof.* By Lemma D.1, for any palyer $i \in [m]$, we have

$$\text{swap-regret}_T \leq \frac{n \log n}{\eta} + 2\eta \sum_{t=2}^{T} \|x_i^t - x_i^{t-1}\|_1^2 + 2\eta \sum_{t=2}^{T} \|\ell_i^t - \ell_i^{t-1}\|_\infty^2$$

$$\leq \frac{n \log n}{\eta} + 2\eta \sum_{t=2}^{T} \|x^t - x^{t-1}\|_1^2 + 2m\eta \sum_{t=2}^{T} \sum_{j \neq i} \|x_j^t - x_j^{t-1}\|_1^2$$

514    where $w^t$ denotes the other player's strategy. Moreover, since $Q^{(t-1)}$ is $(6\eta, \ldots, 6\eta)$ approximates
515    to $Q^{(t)}$, we know

$$\|x_i^t - x_i^{t-1}\|_1 \leq 8 \cdot \sum_{i=1}^{n} 6\eta = O(n\eta)$$

516    holds for any $i$. Thus we have

$$\text{swap-regret}_T \leq \frac{n \log n}{\eta} + 2\eta \sum_{t=2}^{T} \|x^t - x^{t-1}\|_1^2 + 2m\eta \sum_{t=2}^{T} \sum_{j \neq i} \|x_j^t - x_j^{t-1}\|_1^2$$

$$\leq \frac{n \log n}{\eta} + O(\eta^3 n^2 m^2 T).$$

517    Choosing $\eta = O\left((\frac{\log n}{nm^2 T})^{1/4}\right)$, the regret is

$$\text{swap-regret}_T = O\left(n^{5/4} (\log n)^{3/4} T^{1/4} m^{1/2}\right).$$

518    □

# E    Price of anarchy

520    In this section, we show that a large class of no swap regret algorithm satisfies the *low approximate*
521    *regret* property (see Definition E.2). Thus when all players adopt such algorithm, they experience fast
522    convergence to an approximately optimal social welfare in *smooth games* (see Definition E.1). In
523    particular, we show that the average social welfare converges to an approximately optimal welfare
524    at rate $O(1/T)$. The proof in this section is straightforward, our aim is to point out that such fast
525    convergence rate generally holds for no-swap regret algorithms. We first introduce the smooth game.
526    Recall $\mathcal{L}(\mathbf{x}) = \sum_{i \in [m]} \mathcal{L}_i(\mathbf{x})$ is the summation of each individual's loss under strategy profile $\mathbf{x}$.

527    **Definition E.1** (Smooth game). *A cost minimization game is $(\lambda, \mu)$-smooth if for all strategy profiles*
528    $\mathbf{x}$ *and* $\mathbf{x}^\star$, $\sum_i \mathcal{L}_i(x_i^\star, x_{-i}) \leq \lambda \cdot \mathcal{L}(\mathbf{x}^\star) + \mu \cdot \mathcal{L}(\mathbf{x})$.

529    A wide range of games belongs to smooth game, including routing games, auctions, etc. We refer
530    interested reader to [25] for detailed coverage.

531    We next introduce the definition of low approximate regret.

**Definition E.2** (Low approximate regret [14])**.** *A learning algorithm satisfies the low approximate regret property for given parameters $(\epsilon, A(n))$, if*

$$(1 - \epsilon) \sum_{t=1}^{T} \langle x^t, \ell^t \rangle \leq \min_i L(i) + \frac{A(n)}{\epsilon}.$$

**Lemma E.3.** *The BM reduction transfers the low approximate regret property. In particular, if we reduce from a no external regret algorithm satisfying low approximate regret with $(\epsilon, A(n))$, then the no swap regret algorithm satisfies low approximate regret with $(\epsilon, nA(n))$.*

*Proof.* For any fixed $i$, using the low approximate regret property, we know

$$(1 - \epsilon) \sum_{t=1}^{T} \langle q_j^t, x^t(j)\ell_t \rangle \leq \min_{i'} \sum_{t=1}^{T} x^t(j)\ell^t(i') + \frac{A(n)}{\epsilon} \leq \sum_{t=1}^{T} x^t(j)\ell_t(i) + \frac{A(n)}{\epsilon}.$$

Consequently, we have

$$
\begin{aligned}
(1 - \epsilon) \sum_{t=1}^{T} \langle x^t, \ell^t \rangle &= (1 - \epsilon) \sum_{t=1}^{T} \langle x^t Q^{(t)}, \ell^t \rangle \\
&= (1 - \epsilon) \sum_{t=1}^{T} \sum_{j=1}^{n} \langle x^t(j) q_j^t, \ell^t \rangle \\
&= (1 - \epsilon) \sum_{j=1}^{n} \sum_{t=1}^{T} \langle q_j^t, x^t(j)\ell^t \rangle \\
&\leq \sum_{j=1}^{n} \left( \sum_{t=1}^{T} x^t(j)\ell^t(i) + \frac{A(n)}{\epsilon} \right) \\
&= \sum_{t=1}^{T} \sum_{j=1}^{n} x^t(j)\ell^t(i) + \frac{nA(n)}{\epsilon} \\
&= \sum_{t=1}^{T} \ell^t(i) + \frac{nA(n)}{\epsilon}.
\end{aligned}
$$

Thus concluding the proof. $\qquad\square$

A direct corollary of Lemma E.3 and Theorem 3 in [14] is

**Theorem E.4.** *In a $(\lambda, \mu)$-smooth game, if all players use no swap regret algorithm generated from BM reduction and a no external regret algorithm satisfying low approximate regret property with parameter $\epsilon$ and $A(n) = \log n$, then we have*

$$\frac{1}{T} \sum_{t=1}^{T} \mathcal{L}(\mathbf{x}_t) \leq \frac{\lambda}{1 - \mu - \epsilon} \cdot \mathrm{OPT} + \frac{m}{T} \cdot \frac{1}{1 - \mu - \epsilon} \cdot \frac{n \log n}{\epsilon}.$$

*where* $\mathrm{OPT}$ *denotes the optimal social welfare, i.e.,* $\min_{\mathbf{x}} \mathcal{L}(\mathbf{x})$.