[Reviews · NeurIPS 2020]

Review 1

Summary and Contributions: The paper considers Hedge and optimistic Hedge in the context of repeated games. Offers an improved T^{1/6} regret bound of optimistic Hedge in two player games if both players play optimistic hedge, a T^{1/2} lower bound for regular Hedge in the same context, and a T^{1/4} bound for optimistic Hedge for swap-regret via the reduction of Blum-Mansour.

Strengths: The improved bound for optimistic Hedge in 2-player games is very interesting, and extending this line of work to swap regret is also interesting. I find the lower bound for regular Hedge not so surprising, but good to know

Weaknesses: It is unfortunate that the improved regret bound for Optimistic Hedge only applies to two-player games

Correctness: I think so

Clarity: yes

Relation to Prior Work: yes

Reproducibility: Yes

Additional Feedback:


Review 2

Summary and Contributions: Standard low-regret algorithms guarantee O(sqrt(T)) regret after T rounds (which is tight). One common application of low-regret algorithms is to play an n-action game (in settings like this, it is known that if all players are running low-regret algorithms, then their empirical strategies will converge to specific types of equilibria for the game). It was shown in a series of works that in this more structured setting, it is possible to design algorithms with better regret guarantees; in particular, Syrgkanis et al show that an algorithm known as “Optimistic Hedge” (a generalization of the standard “hedge” / multiplicative weights algorithm) achieves regret bounds on the order of O(T^{1/4}) when players in 2 player games both play it. This paper examines the Optimistic Hedge algorithm in further detail, significantly improving the bounds shown by Syrgkanis et al. Specifically, this paper: 1. Shows that if both players in a two-player game run Optimistic Hedge, each player’s regret is at most O~(T^{1/6}). 2. Shows that if both players in a two-player game both run standard hedge, there are cases where they must incur Omega(sqrt(T)) regret. 3. Shows that if Optimistic Hedge is used as the input to the blackbox swap-regret construction of Blum and Mansour, this leads to a swap-regret algorithm with total regret at most O~(sqrt(m)*T^{1/4}) when m players all use this algorithm to play in a game (in contrast with the bound of O~(sqrt(T)) for the swap-regret from the standard construction). These improvements rely on the fact that Optimistic Hedge converges quickly if the loss vectors and strategies it outputs are relatively stable. To prove (1), the authors show that this is indeed the case when both players run Optimistic Hedge in a 2 player game (by carefully relating the losses of one player to the strategies of the other and vice versa). To prove (3), the authors combine these observations with a careful analysis of how this impacts the Markov chain’s stationary distribution in the Blum Mansour reduction. Via the matrix-tree theorem, they show that small multiplicative changes in the edge weights of the Markov chain (which are guaranteed by this property of Optimistic hedge) lead to small L1 changes in the Markov chain’s stationary distribution, which in turn leads to better swap regret bounds.

Strengths: - Makes significant progress on a number of open problems of Syrgkanis et al.: 1. they improve the best known convergence rate for two-player games from O(T^{-3/4}) to O(T^{-5/6}), 2. they prove a lower bound of O(T^{-1/2}) on the convergence rate of vanilla Hedge (matching best known analysis). - They extend these results to prove faster convergence rates for swap-regret algorithms (which guarantee convergence to correlated equilibria). - Proof techniques (e.g. analysis of the Markov chain of the Blum and Mansour reduction) might be of independent interest.

Weaknesses: - Can you say anything about the convergence rate in m-player games if everyone runs Optimistic Hedge? Is it possible to improve upon the O(T^{-3/4}) convergence rate of Syrgkanis et al.

Correctness: I have not checked the proofs in the Appendix, but all proofs in the main paper appear correct conditional on this (and the methods seem plausible).

Clarity: The paper was very well-written and easy to understand.

Relation to Prior Work: This paper cites the relevant prior work.

Reproducibility: Yes

Additional Feedback: This is an excellent paper that makes significant progress on a number of interesting problems. Barring any issues with the proofs, I strongly recommend acceptance.


Review 3

Summary and Contributions: The paper analyzes both optimistic and regular hedge in a multi-player setting. * For two players they improve the convergence bound to 1/T^{5/6} (from 1/T^{3/4}) using optimistic hedge * For standard Hedge they derive a lower bound of 1/sqrt{T} (which hold for penny-matching, in the interesting regime of the learning rate) * For m players they improve the bound of the swap regret (using the reduction of BM and the optimistic Hedge)

Strengths: All the results results are very interesting, and some resolve open problems from previous work; * For two players they improve the convergence bound to 1/T^{5/6} (from 1/T^{3/4}) using optimistic hedge. This is a very fundamental result! * For standard Hedge they derive a lower bound of 1/sqrt{T} (which hold for penny-matching, in the interesting regime of the learning rate) This shows a clear separation between the standard and the optimistic versions! * For m players they improve the bound of the swap regret (using the reduction of BM and the optimistic Hedge) This is a surprising result in my view.

Weaknesses: As usual there is more that case be done, and the authors cover it very well in their discussion section.

Correctness: Looks good to me (did not verify the proof in the supplementary material, but the big picture is clear)

Clarity: Very well written!

Relation to Prior Work: Very nicely done.

Reproducibility: Yes

Additional Feedback: Following the rebuttal I did not change my score

[Author Response · NeurIPS 2020]

We thank all reviewers for their insightful comments.

We left as an open problem in the paper for extending the convergence result to general multiplayer games when everyone runs Optimistic Hedge. Our current analysis does not work for three-player games due to some technical difficulties. But we believe that some of the technical ingredients and insights developed in this work could be useful for future research on this problem.

[Meta-Review · NeurIPS 2020]

All reviewers support accept strongly. Clear accept.